

# Sensitivity of air quality modelling to different emission inventories: a case study over Europe

Philippe Thunis, Monica Crippa, Cornelis Cuvelier, Diego Guizzardi, Alexander de Meij, Gabriel Oreggioni, Enrico Pisoni

European Commission, Joint Research Centre (JRC), Ispra, Italy

*Correspondence to*: E. Pisoni (enrico.pisoni@ec.europa.eu)

**Abstract.** In this paper we run the EMEP air quality model with 3 different emission inventories at 2015, that is to say using CAMS, EMEP emissions, and EDGAR. The EMEP model has been run for the entire year 2015, and resulting concentration

results have been compared with 'background' monitoring stations. Results show that the air quality model, run with the 3 emission inventories, provide similar results despite the emission differences. More in details, EDGAR is providing slightly better validation results for PM2.5, while the EMEP emissions are slightly better to model yearly average $NO_2$. The main differences among the model applications arise in the Eastern part of Europe, where the values between the officially estimated emissions and those independently estimated by EDGAR are higher. Results suggest that EDGAR, despite being a

methodology aimed at global coverage, with independent sources for activity level, technologies and emission factors and generic gridding practices, can be effectively used for air quality modelling in Europe. The EDGAR dataset (v5.0) used in this paper is available at: https://data.europa.eu/doi/10.2904/JRC_DATASET_EDGAR (link: EDGAR v5.0 Global Air Pollutant Emissions, Crippa et al., 2020a).

## 1 Introduction

Air quality modelling is a key tool to inform policymaking, because it represents the only available methodology to provide insights on (1) the potential impacts of emission abatement measures on air quality and (2) concentration levels reached over territories where no monitoring station are operating (EEA, 2019). While models are continuously being improved (e.g. better representation of the atmospheric chemical processes, finer grid resolution to capture explicitly smaller scale features, etc.), they also keep depending on specific inputs. Emission inventories are generally identified as the key input to the air

quality modelling chain, and many studies point to emission inventories as the most uncertain factor among the different components of air quality models (e.g. meteorology, boundary conditions, model parameters) (Russell and Dennis, 2000; Davison et al., 2011; Viaene et al., 2016, Pisoni et al., 2018). This uncertainty can be especially large for some activity sectors due to the lack of knowledge on the activities producing emissions (e.g. agriculture, waste, etc.) and/or on the technological and abatement measures penetration influencing the identification of adequate emission factors. In their inter-





comparison of six aggregated top-down inventories, Trombetti et al. (2018) showed that uncertainties could reach more than 100% for some sectors. Representative emission inventories are hence crucial to air quality applications as they will determine to a large extent the accuracy of the subsequent air quality modelling results (Georgiou et al., 2020).

In literature, numerous studies have assessed the sensitivity of this particular model input on the modelled concentrations and related indicators. Zhu et al. (2019) simulated the changes in ozone and fine particulate matter (PM2.5) due to an improved

estimation of VOC emissions in California. Results showed that simulated daily maximum 8-hr ozone concentrations could increase by 17.4 ppb in summer and by 15.6 ppb in winter, and the 24-hr maximum PM2.5 could increase by 7.8 μg/m$^3$ in winter. In another paper, and from a global perspective (Crippa et al., 2019), the authors investigated emission inventory uncertainties and their propagation to PM2.5 concentrations. They estimated 2.1 million premature deaths per year due to PM2.5 concentrations, with an uncertainty (due to emission variability) of more than 1 million premature deaths per year.

In any case, because of the continuous evolution of the air quality models but also of emission inventories, it is important to repeat these sensitivity analysis, to understand how recent updates have changed our representation and our understanding of the air quality processes. With this work, we contribute to this process. In particular, we use the EMEP regulatory air quality model (Simpson et al., 2012) and feed it with three EU wide emission inventories: EMEP own emissions, CAMS and EDGAR. We use here (and in the rest of the paper) the 'EMEP' label to refer to the emissions as used in the EMEP model

for their policy applications (EMEP, 2019; see https://www.ceip.at/ms/ceip_home1/ceip_home/webdab_emepdatabase/emissions_emepmodels/), 'CAMS' to refer to the recently released v2.2.1, as part of the Copernicus Atmospheric Services (Kuenen et al., 2014; https://atmosphere.copernicus.eu/); while EDGAR refers to the recently released v5.0 (Crippa et al., 2018; https://edgar.jrc.ec.europa.eu/overview.php?v=50_AP). In Section 2, we briefly describe the modelling tools and data

applied to perform the simulations, as well as the tools used to analyse the results; we also present the main features of the recently released EDGAR emission inventory, stressing key differences with EDGAR previous versions. In Section 3, we compare the emission inventories and identify the main differences among them, while the resulting modelled concentrations are discussed in Section 4 and Section 5 (with validation against observations, and focusing on PM2.5, PM10, NO$_2$ and O$_3$). Finally, in Section 6, we discuss the main assumptions and limitations of the approach.

## 2 Modelling setup


In this section we briefly describe the EMEP air quality model applied for the simulations and the three emission inventories used. Only the main properties of the emission inventories and air quality model applied are detailed here (for more information, we suggest the reader to look at the related references).



## 2.1 Air Quality Model

The EMEP model (Simpson et al., 2012) version rv_33 (https://github.com/metno/emep-ctm) is used over continental Europe to study the sensitivity of different emission inventories on calculated gas and PM2.5/PM10 concentrations. The domain stretches from -15.05° to 36.95° longitude and 30.05° to 71.45° latitude with a horizontal resolution of 0.1° x 0.1° longitude/latitude. The model has 20 vertical levels, with the first level at around 7 m. The model uses meteorological initial and lateral boundary conditions from the European Centre for Medium Range Weather Forecasting Integrated Forecasting

System (ECMWF-IFS). The considered meteorological year is 2015. Detailed information on the meteorological driver, land cover, model physics are described in Simpson et al. (2012) and in the EMEP Status Report 2017 (EMEP, 2017).

## 2.2 Emission inventories

In terms of emission inventories, we test 3 'candidates' as input to the EMEP air quality model: EDGAR, EMEP and CAMS-REG-AP emissions. In the next sections of the manuscript, both a comparison of the emissions themselves and their

impact on concentrations (as produced by the EMEP model based on the different emissions) are presented. In this section, on the contrary, we briefly detail the key features of the three emission inventories.

The EDGAR database (Emissions Database for Global Atmospheric Research, Crippa et al., 2018) is a bottom-up global database providing historic emission time series and grid maps for all countries from 1970 until 2015, for both air pollutants and greenhouse gases, calculated in a consistent and transparent way and therefore allowing comparability amongst

countries. EDGAR incorporates a full differentiation of emission processes with technology-specific emission factors and additional end-of-pipe abatement measures.

The EMEP emissions (Mareckova et al., 2017) are compiled within the "UNECE co-operative programme for monitoring and evaluation of the long-range transmission of air pollutants in Europe" (unofficially 'European Monitoring and Evaluation Programme', EMEP). EMEP is a scientifically based and policy driven programme under the Convention on Long-range

Transboundary Air Pollution (CLRTAP) for international co-operation, that has the final aim of solving transboundary air pollution problems. More specifically, the EMEP emissions are built from officially reported data provided to CEIP (Centre of Emission Inventory and Projection, a body of EMEP) by the Convention Parties (Member States, in Europe); emissions are gap-filled with gridded TNO data from CAMS and EDGAR (upgraded by point source information available under E-PRTR), if needed, for use in the EMEP air quality model.

The CAMS-REG-AP (CAMS regional anthropogenic emission inventory, Granier et al, 2019) covers emissions for the UNECE-Europe for the main air pollutants and greenhouse gases. The CAMS-REG-AP methodology starts from the emissions reported by European countries to UNFCCC (for greenhouse gases) and to EMEP/CEIP (for air pollutants), aggregated into different combinations of sectors and fuels. Then, these emissions are gridded using ad-hoc proxies, that differ from the ones used in the EMEP emissions.

A summary of the information for the three considered emission inventories is provided in Table 1.



**Table 1: Overview of the main characteristics for the three emission inventories.**

| | EDGAR | EMEP | CAMS-REG-AP |
|---|---|---|---|
| **TIME COVERAGE** | 1970-2015 | 1990-2017 | 2000-2016 |
| **SPATIAL COVERAGE** | World | Europe | Europe |
| **SPATIAL RESOLUTION** | 0.1 x 0.1 deg | 0.1 x 0.1 deg | 0.05 x 0.1 deg |
| **TEMPORAL RESOLUTION** | Monthly | Yearly | Yearly |
| **POLLUTANTS** | CH4, CO2, N20, NMVOC, CO, SO2, NOx, NH3, PM10, BC, OC, PM2.5 | NMVOC, CO, Nox, NH3, TPM, PM10, PM2.5, Hg, Cd, Pb | CH4, NMVOC, CO, SO2, NOx, NH3, PM10, PM2.5 |
| **METHODOLOGY** | Independent activity and technology estimation, own gridding | Official emission estimation and gridding (when available) | Official emission estimation and own gridding |
| **REFERENCES** | Crippa et al., 2018 | Mareckova et al., 2017 | Granier et al., 2019 |
| **LINK TO DOWNLOAD** | https://data.europa.eu/doi/10.2904/JRC_DATASET_EDGAR | https://www.ceip.at/the-emep-grid/gridded-emissions | https://eccad.aeris-data.fr/catalogue/ |



## 3. EDGAR v5.0 and its comparison with other emission inventories

As mentioned above, we consider 3 emission inventories as input to the EMEP air quality model: 'EDGAR', 'EMEP' and
'CAMS' emissions. While we refer to other publications for details on EMEP and CAMS (i.e. see EMEP, 2010; Mareckova
et al., 2017; Granier et al, 2019) we focus here on the most recent update of the EDGAR, that is now in its version 5.0. The
Emissions Database for Global Atmospheric Research (EDGAR) is a global inventory providing greenhouse gas and air
pollutant emissions estimates for all countries from 1970 till now, covering all IPCC reporting categories. EDGAR emissions
are computed using a consistent methodology across countries, making use of international statistics for activity data (e.g.
energy balances from IEA, FAO data from agriculture, USGS for clinker and mineral production, IFA for fertiliser
production). If required, this activity data is technologically disaggregated, for example for energy industries and road
transport, using the most up to date information from installed capacity such as the UDI Platts (S & P, 2018) database and
fleet distribution data from the 'Emisia' company (EMISIA, 2018).  When possible, regional or country based technological
Tier 2[1] emission factors are used.  EDGAR methodology is further described in Janssens-Maenhout et al., 2019, Oreggioni et
al., 2020a, 2020b. EDGAR also provides spatially distributed data with 0.1 x 0.1 degree resolution and temporarily
disaggregate emissions down to hourly values (Crippa et al. 2020b).  EDGAR's completeness, time coverage and robust
methodology has allowed EDGAR to be a benchmark for the emission inventory scientific community, being also a useful
tool to complete the global picture in terms of global carbon budget and air pollutant emissions, enabling its use for the
monitoring of progress of abatement measures and the identification of sector of concern (Oreggioni et al., 2020a). In this
work, we use the latest EDGAR air pollutant dataset (EDGARv5.0) which includes several updates compared to the former
releases (Crippa et al. 2018):

- new spatial proxies to distribute population-related emissions based on the Global Human Settlements Layer
product (Pesaresi et al., 2019; Crippa et al., in prep.);
- updates in the technologies, emission factors and end-of-pipe reductions for the power generation sector (for the 27
Members of the European Union, UK and China) based on the UDI  Platts database 2018 release, and the
implementation of the regulation taking place in these region (Oreggioni et al., 2020b);
- updates in the technologies, emission factors and end-of-pipe reductions for the road transport sector ((EU27+UK).
These updates are based on the EMISIA 2018 data (Emisia 2018);
- Estimates of particulate matter emissions from road surface wear and road vehicle tyre and break wear, based on the
EMEP/EEA guidebook 2019 (EMEP/EEA, 2019) Tier 1 approach;
- new high resolution temporal profiles, although not used in this work to keep consistent temporal disaggregation
within all inventories (Crippa et al., 2020b).

---

[1] For Tier 2 definition, see EMEP/EEA air pollutant emission inventory guidebook 2019.



## 3.1 Comparison strategy

Comparing emission inventories is a challenging task as many dimensions are involved on activity sectors, technological
information, implementation of abatement measures, geographical coverage, spatial disaggregation of the emissions,
pollutants, etc. In the following sections, we structure the comparison geographically and address first a comparison at the
EU-wide and country scales before zooming in at the regional scales. For each of these geographical scales, we then analyse
how consistent the three inventories are (1) in terms of their EU sectorial share (i.e. for a given pollutant and a given sector,
do the inventories distribute similarly the emissions across the EU countries?) and (2) in terms of their national share (i.e. for
a given country and a given pollutant, do the inventories distribute similarly the emissions across the different sectors?).  To
synthetize the information, we use the EMEP inventory as reference and compare the correlations obtained between CAMS
and EMEP with those obtained between EDGAR and EMEP. In the context of emission comparison, the use of EMEP as
reference is meant as a reference point but has no implication on the quality of the inventory itself, as we do not know the
'true' emission value.

In the GNFR (Gridded Nomenclature for Reporting) classification, used for the 3 inventories, emissions are initially
categorized into 13 sectors: power plants, industrial facilities, other combustion, fugitive emissions, solvents, road transport,
shipping, aviation, off-road transport, waste, agriculture livestock, other agriculture and other. For convenience, we
aggregated these emissions into 8 sectors: (1) Industry (first three original sectors), (2) Fugitive and solvents, (3) Road
transport, (4) Shipping, (5) Aviation, (6) Off-road, (7) Agriculture (sum of two original agriculture related sectors) and (8)
Other (containing the remaining sectors).

## 3.2 Comparison at EU scale

We first compare the three inventories from a continental perspective (considering the countries in a domain that span from
15.05° to 36.95° longitude and 30.05° to 71.45° latitude[2]). In terms of totals over all covered countries (Figure 1), all
inventories agree well among themselves with the exception of VOC for which EDGAR provides a larger estimate and
CAMS a smaller estimate, in comparison to EMEP.

---

[2] In particular, these are the countries considered (using international labelling system): AT, BE, BG, CH, CY, CZ, DE, DK,
EE, EL, ES, FI, FR, HR, HU, IE, IS, IT, LI, LT, LU, LV, ME, MK, MT, NL, NO, PL, PT, RO, SE, SI, SK, TR, UK.



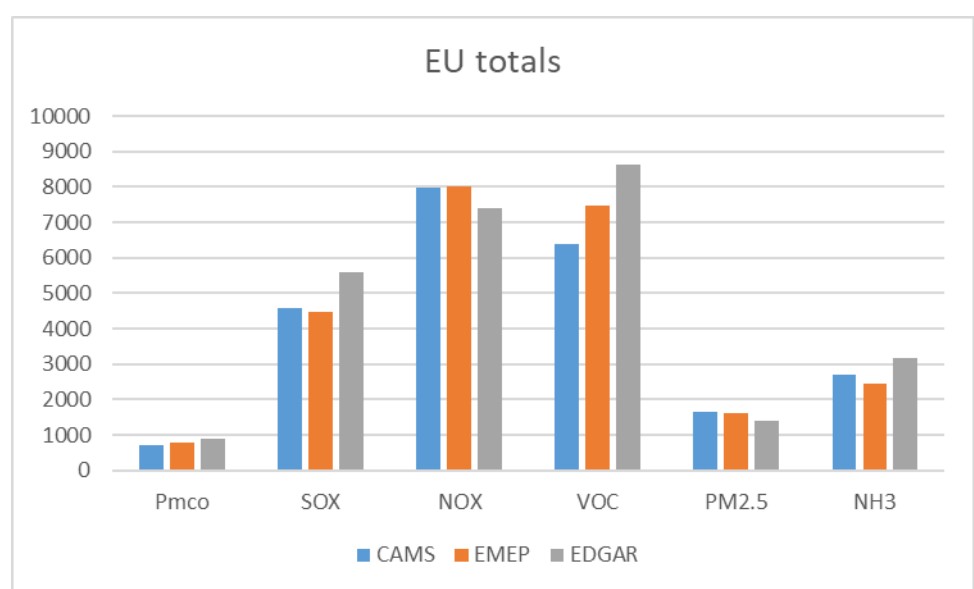

**Figure 1: Comparison of the total emissions for the three emission inventories (unit: kton/year).**

If we look at how the emissions are distributed among the countries for each pollutant, Figure 2 (top-right) compares the CAMS-EMEP correlation with the EDGAR-EMEP correlation for the 6 pollutants considered. For this comparison,

emissions have been summed up sector-wise. The figure therefore provides information on the level of agreement between the distribution of total emissions among the EU countries. We note that both correlations are very high (larger than 85%) with the exception of the PM coarse fraction ('PMco', the part of PM between PM2.5 and PM10) for which EMEP agrees well with CAMS but not with EDGAR. The sectorial details (Figure 2 top-left) indicate that this issue with 'PMco' mostly originates from the industrial and agricultural sectors. NOx emissions from shipping and NH3 emissions from the industrial

sector also show quite low correlations. This can be explained by the fact that the different inventories have different level of emission completeness for the international components for shipping and aviation. EDGAR, for example, do not include international shipping and aviation in total national emissions because it uses IEA Fuel Balances (IEA, 2017) as data source for fuel consumption and fuel burned in international shipping and aviation are supplied at total global levels.

Obviously, the correlations between the country shares of the different inventories are influenced by the size of the countries,

and even more by its population. The differences in terms of population and associated emissions drive the correlation coefficients. To prevent this problem, we provide the same emission comparison, but per capita (Figure 2 bottom). Correlations drop significantly with the exception of NH3 and PM2.5. Differences generally become larger between EMEP and EDGAR than between EMEP and CAMS, reflecting the independent process followed to generate the EDGAR inventory, and therefore the use of independent procedures to estimate the EDGAR emissions.

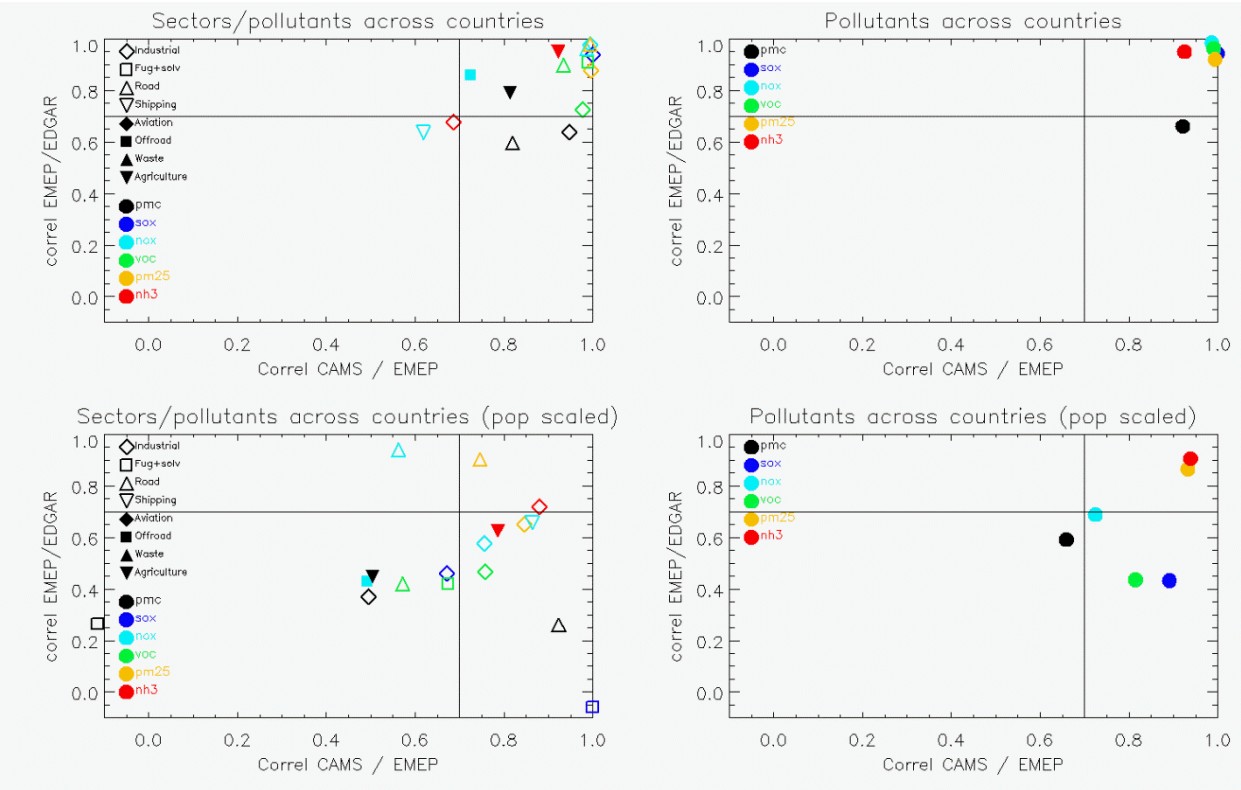


**Figure 2: (top-right) comparison between the EMEP-CAMS and EMEP-EDGAR correlations for each pollutant. The correlations indicate how coherent the distribution of the emissions are among countries for two inventories. Emissions are summed up sector wise. (top-left) same as top-right but with the sectorial details. Note that only the sectors that represent at least 5% of the total emissions for one pollutant are displayed. (Bottom) same as top figures but country emissions are normalized by capita.**

**3.3 Comparison at country scale**

Figure 3 provides a breakdown of the emissions in terms of countries. Both biases and correlations are provided. The correlation informs on how well models agree with the allocation of emissions among different sectors for a given country and sector. From Figure 3, we note the following points:

- All inventories agree well for NOx and PM2.5 emissions for which the bias is limited (Figure 3 top-right);

- For the coarse fraction of PM and SO2 (Figure 3 bottom-right), the CAMS and EMEP inventories agree well but differ from the EDGAR estimates, that shows larger emission in several countries. For PM coarse (PMco), this bias is mostly occurring in Eastern countries but countries such as Italy or Spain are also concerned, being these differences mostly coming from small combustion sources. Combustion and end of pipe technologies are currently in the process of being updated in EDGAR, being the methodology for emission quantification reviewed, as shown

in Muntean et al (2020);

- Differences for $SO_x$ between EDGAR and other inventories were also observed, especially in Eastern European countries. Main sources for $SO_x$ emissions are power and heat plants, especially those ones fuelled with coal. Two





reasons can explain the observed differences. On one hand, EDGAR $SO_x$ emissions for this sector are calculated using a capacity and regulatory based methodology, considering the power plant inventory in UDI Platts database (S&P, 2018). It may be the case that this data source overestimates the shares of plants with low thermal capacities, for example not considering that they may have been retired. Smaller thermal capacity units are required to fulfil with less strict emission limits thus being equipped with lower removal rate desulfurization processes, consequently leading to higher $SO_x$ emissions. On the other hand, in EDGAR, plants are all assumed to respect the limits of the 2001 Large Combustion Plant Directive, even if this legislation was updated in 2016. Plants are currently in the process of implementing the new regulatory changes and this could not still well captured in EDGAR;

- For VOC (Figure 3 bottom-left), the EDGAR overestimation (with respect to EMEP) is the largest in countries like Belgium, Austria, Switzerland, Germany or Finland while for NH3, it is the largest in Austria, Belgium, Denmark, the Netherlands, Poland or Sweden. In the case of VOC emissions, the differences take place in industrial combustion sources and fugitive emissions, for which natural gas play a key role; this shows that VOC emissions from natural gas in EDGAR are higher than in the other inventories;

In terms of correlation (Figure 3 top-left), i.e. in terms of consistency between the country sectorial share in two inventories, the comparison shows again a more consistent picture between CAMS and EMEP than between EMEP and EDGAR. The largest differences occur for PMco and VOC. The differences in the sectorial allocation can be important. If we take Denmark as an example, CAMS distributes VOC in equal shares on the industrial, shipping and off-road, EMEP mostly allocates it to industry and off-road (but in a lesser proportion), while EDGAR puts two thirds of emissions into off-road and the remaining to the fugitive and solvents sectors. It is important to stress that these sectorial emission differences could impact the modelling of the current atmospheric levels (because sectors are distributed differently geographically) but could also impact the modelling of emission abatement measures that are directly attached to specific sectors.

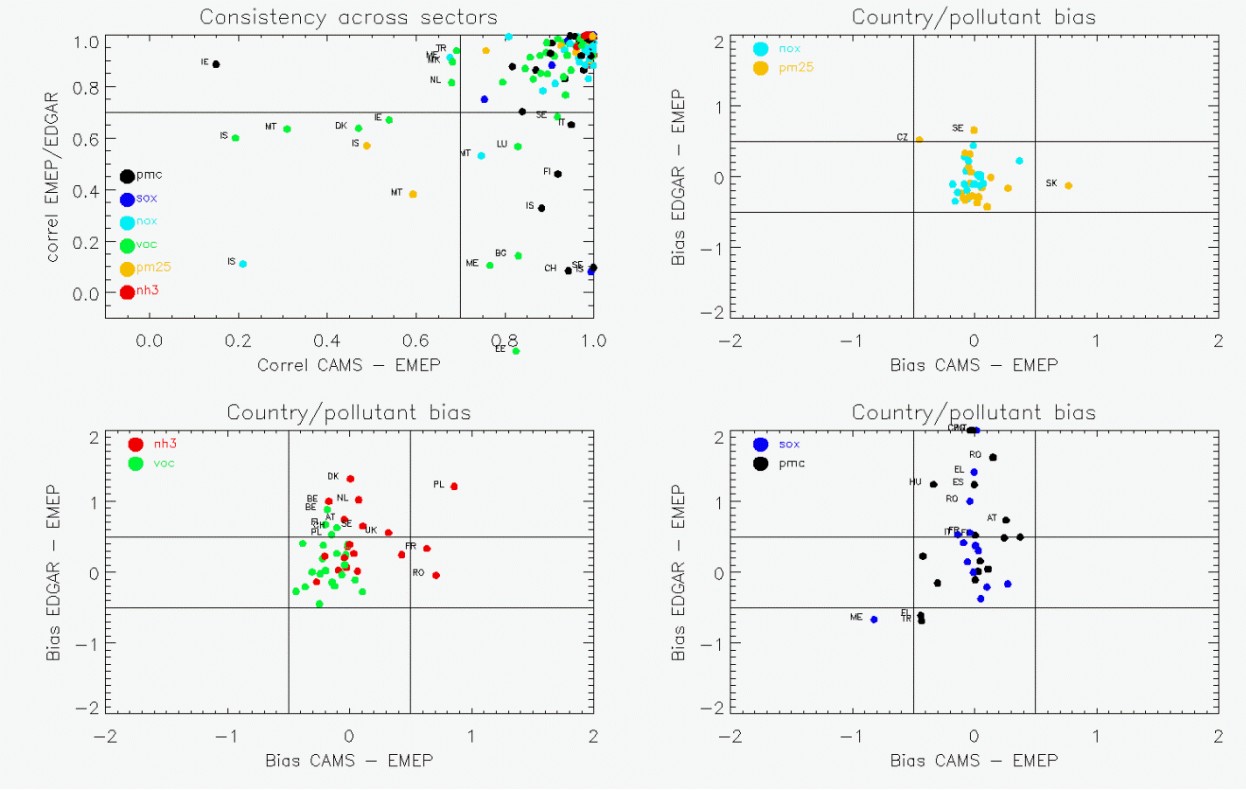

**Figure 3: (top-left) comparison between the EMEP-CAMS and EMEP-EDGAR correlations for all pollutants and country. The correlations indicate how coherent the distribution of the emissions are among sectors for each country between two inventories. (top-right) PM2.5 and NOx comparison between the EMEP-CAMS and EMEP-EDGAR biases for each country. (Bottom) same as top right figure but for NH3, VOC, PMco and SOx.**

At the country level, more important differences are seen between EDGAR and the other two inventories, while CAMS and EMEP do have a higher level of agreement. Results are quite similar for PM2.5 and NOx, but larger differences are found for the other compounds. It is important to note that these differences are not generalized geographically and they will therefore lead to different pollutant ratios (e.g. NOx/VOC) in some countries and not in others. These differences might affect the chemical regimes and therefore the concentrations in the air, as explained in Section 4 and Section 5. In the following section, we analyse in more details the situation in three regional areas where the reported pollutant concentrations generally exceed the EU thresholds: the so-called 'black triangle', the Benelux and the Po-Valley in Northern Italy.

### 3.4 Comparison at regional and city scales

In this section, we compare the inventories in three EU regions that regularly suffer from high pollution levels. The three regional domains: the Po Valley, the "black triangle" (covering the southern part of Poland, Eastern Part of Czech Republic and Slovakia), and the Benelux are depicted in Figure 4 (top left). The following points can be highlighted:




• With the exception of PM coarse (PMco), the CAMS and EMEP inventories agree relatively well for all pollutants (orange columns close to 1). For PMco, CAMS largely underestimate the values with regards to EMEP;

• Differences between inventories are much larger over the "black triangle" area than over the two other regions. This is visible for all pollutants, excepted NOx and SOx.

For these three regions, we also analyse the emission share between urban and rural emissions. We fix a minimum (arbitrary)
threshold of 300 inhabitants[3] per km2 to define urban cells and calculate the emissions percentage allocated to these urban cells. This is depicted by the circles on top of the bars in Figure 4 that provide a quantification of the ratio of the urban emission between inventories (a value of 1.2 for the blue circle will indicate that the CAMS inventory allocates 20% more emissions to the urban areas than the EMEP inventory). For the Benelux and the Po-Valley, all three inventories allocate the emissions similarly while major differences are observed over the black triangle area. EDGAR and CAMS in a lesser
measure allocate much more emissions to the urban areas than EMEP. For EDGAR, the differences reach or exceed 50% for VOC and PM. This is likely to have an impact on the modelled concentrations as many stations are located in or around urban areas.

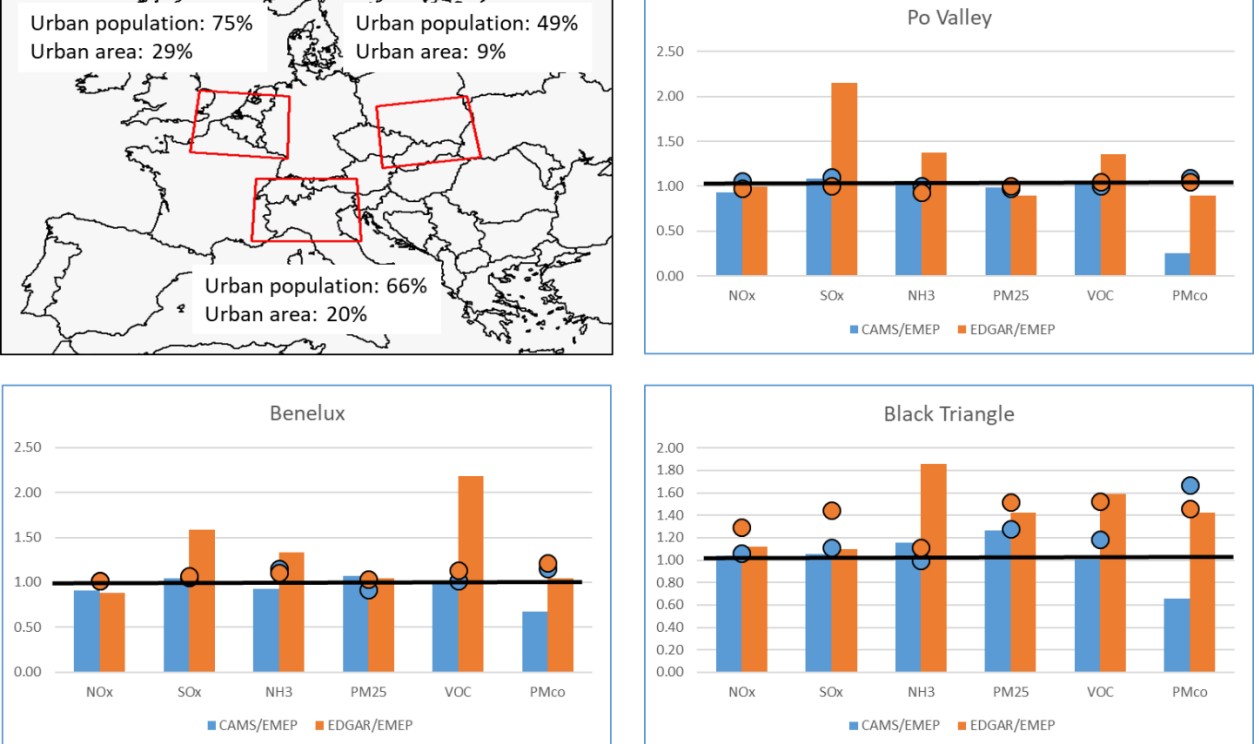

**Figure 4: Comparison of the emission totals for the three regional domains depicted in the top-left panel. Values are given as ratios**
**between the total emissions over the domain between the CAMS and EMEP (blue columns) and EDGAR and EMEP (orange**

---

[3] For the choice of this value, please refer to Eurostat: https://ec.europa.eu/eurostat/statistics-explained/index.php?title=Archive:Urban-rural_typology&oldid=78848.





**columns) inventories. The orange and blue bullets correspond to the same ratios but for the share of emissions allocated to the urban areas (see text for details).**

## 4. Comparison of the PM, O₃ and NO₂ concentration

Feeding the EMEP model with the three emission inventories (EMEP, CAMS and EDGAR) leads to the spatial yearly mean
concentration fields presented in Figure 5. For $O_3$, the differences among models are minor over the entire domain whereas EDGAR tends to produce larger values than EMEP (differences reach 3 to 5 ug/m3) especially in the Benelux, southern UK, and Paris, but they are also widespread over Germany. These differences are well correlated with lower estimates of the $NO_2$ concentrations in the same regions. Differences are more important for PM, especially for PM2.5. Differences are high (reaching 5 ug/m3) in a wide part of Eastern Europe while they remain minor in other EU regions.

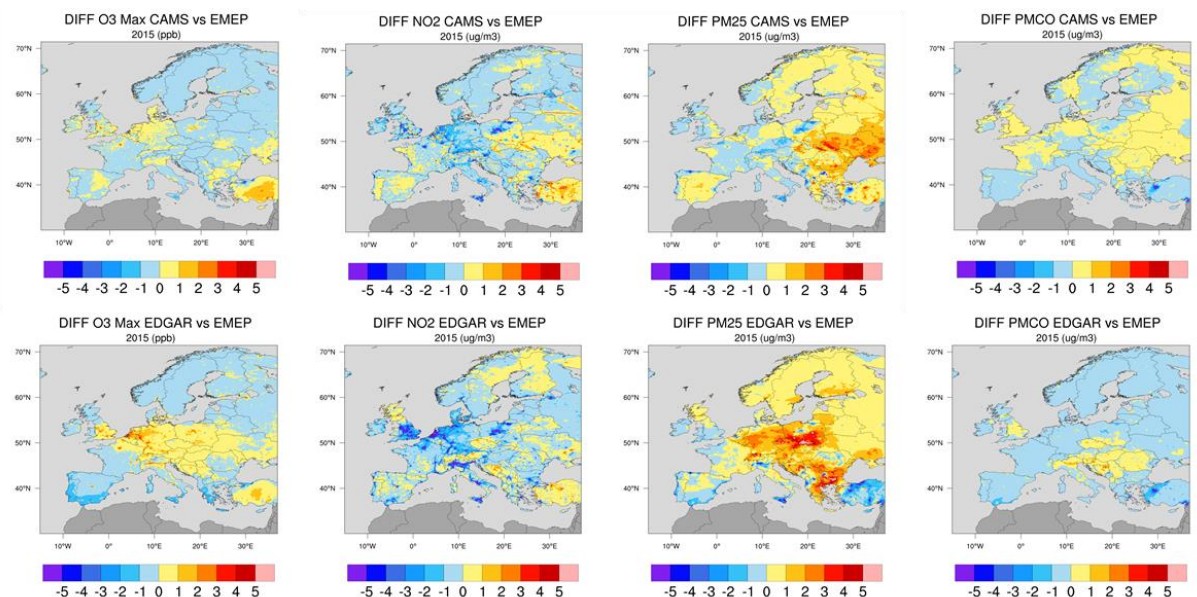


**Figure 5: comparison of yearly averaged concentration fields for O₃, NO₂, PM2.5 and PM coarse between CAMS and EMEP (top row) and EDGAR and EMEP (bottom row). Results are expressed as differences in ug/m3.**

While these differences are significant in some parts of Europe, it is worth to see how they impact the evaluation of the model results with measurements. In the next section, we analyse how different emission input impact the quality of the
modelling results. In particular, we apply specific statistical indicators that inform on their quality for the specific purpose of supporting policy decisions.



## 5. Model validation

### 5.1. Approach

For the evaluation of the model results against measurements, we apply the benchmarking methodology developed in the
frame of the FAIRMODE[4] network (https://fairmode.jrc.ec.europa.eu/). This methodology is primarily based on the calculation of Modelling Quality Indicators (MQI), which compare modelled and measured data, taking the measurement uncertainty into account. The underlying principle of this approach is to consider that measurements are not error-free and allow therefore the model results a margin of tolerance that is proportional to the measurement uncertainty. In other words, the more uncertain the measurement, the larger the tolerance on model results (larger model-observation bias are allowed).
The measurement uncertainty depends on the pollutant and on the concentration level, with larger uncertainties characterising the lower concentration levels. Here, we only briefly summarize the main points of the approach and refer to Thunis et al. (2013a, 2013b) for the complete derivation.

For a single pair of measured-modelled values, the MQI is defined as the ratio of the model (Mi) and measured (Oi) bias to a quantity proportional to the measurement uncertainty, and is calculated as:

$$MQI = \frac{|O_i - M_i|}{\beta U(O_i)} \tag{1}$$


where the index i denotes a given time (hour or day), $U(O_i)$ is the measurement uncertainty and β is a coefficient of proportionality arbitrarily set to 2, thus allowing the deviation between modelled and measured concentrations to be twice the measurement uncertainty in the current formulation. A modelling application is considered to deliver results of sufficient quality when the MQI is less than unity.


Applied to a complete time series, Equation (1) can be generalized to:

$$MQI = \frac{RMSE}{\beta RMS_U} \tag{2}$$

With this formulation, the RMSE (Root Mean Squared Error) between observed and modelled values (numerator) is
compared to a value ($RMS_U$) representative of the maximum allowed measurement uncertainty (denominator). In equation (2), the full expression of $RMS_U$, its simplified parametrization and the necessary input parameters are available in Thunis et al. (2013a).

---

[4] FAIRMODE is a Forum for Air Quality Modelling created for exchanging experience and results from air quality modelling in the context of the Air Quality Directives and for promoting the use of modelling for air quality assessment and management in a harmonized manner between EU Member States.





For yearly averaged pollutant concentrations, the MQI formula is adapted so that the mean bias between modelled and
measured concentrations is normalised by the uncertainty of the mean measured concentration:

$$MQI = \frac{|\bar{O} - \bar{M}|}{\beta U(\bar{O})} \qquad (3)$$

The uncertainty of the averaged concentration $[U(\bar{O})]$ is lower than the uncertainty of an individual measurement $[U(O_i)]$
because it accounts for the compensation of errors due to random noise and other factors like periodic re-calibration of the
instruments, between measurements. The full expression of the uncertainty of the average concentration is also available in
Thunis et al. (2013a).

For the evaluation of the model results against measurements, we apply the following two rules:

- Generally, condition (3) is more stringent than condition (2). The current recommendation of FAIRMODE is that both conditions must be fulfilled;
- Follow the requirements prescribed by the EU Ambient Air Quality Directives (AAQD), equations (2) and (3) must be fulfilled for at least 90% of the available measurement stations.

For the evaluation, we used the Airbase stations as available for 2015 from the EEA website, considering only background
stations with at least 75% data available (we do not consider other type of stations as i.e. traffic stations, as for such
observations this type of model and spatial resolution is not sufficient). The following indicators have been analysed in this
work: daily and yearly averaged $NO_2$, the 8h daily maximum $O_3$ and its average over the summer time (April to October)
and daily and yearly PM10 and PM2.5. In total we used 1764 stations for $NO_2$, 1698 stations for Ozone, 877 stations for
PM2.5 and 1732 stations for PM10.

**5.2. Evaluation at country scale**

Although only one model is used with three inventories, we will refer to three models in the following, for convenience.
These three models correspond to three model configurations: EMEP fed with the CAMS, EDGAR and EMEP inventories,
respectively. We analyse the behaviour of the three models for daily average or daily maximum concentrations as well as for
yearly or summer (for $O_3$) averages and assess their performance with respect to the MQIs defined by equation (2) and (3).
The Target diagrams (Figure 6 to Figure 9) are designed to visualize the daily maximum ($NO_2$), 8 hours daily maximum (for
$O_3$) or daily (PM) MQI and its components. The MQI represents the distance between the origin and a given station
(represented by a point on the diagram). This distance should be less than unity, i.e. fall within the green area for at least
90% of the available stations. In the Target diagram, the X and Y axis correspond to the unbiased root mean square error
($CRMSE$) and to the bias, normalized by the measurement uncertainty, $RMS_U$. For each point on the diagram, the ordinate is
then $BIAS/\beta RMS_U$ and the abscissa $CRMSE/\beta RMS_U$ while the radius is equal to $RMSE/\beta RMS_U$. Because $CRMSE$ is
always positive only the right hand side of the diagram would in theory be needed in the Target plot. The negative X axis





section is then used to provide additional information. When the correlation component dominates the standard deviation component in the CRMSE, a station is represented on the left (and vice-versa). The diagram allows therefore to distinguish stations according to their type of error, whether dominated by bias (either negative or positive), by correlation or by standard deviation.

The MQI for the yearly (PM2.5, PM10 and NO$_2$) or summer (O$_3$) averaged results are generally more challenging to fulfil.

Equation (3) is based on bias only and is used as the main model quality indicator for averaged concentrations. In the scatter plots of Figure 6 to Figure 9, the MQI is used to represent the distance from the 1:1 line. As mentioned above it is expected to be fulfilled (points are in the green area) by at least 90% of the available stations.

To avoid over plotting the entire set of stations on a single diagram, we classify stations into country groups. For all stations within a country group, we then calculate their MQI values, rank them and identify the 90th percentile station which we

represent on the Target and scatter diagrams. The mean bias and mean CRMSE of the stations composing the country group are used to position the point according to the X and Y axis of the Target diagram. For a country group, its right/left location is based on the behaviour of the majority of the stations within that group. If this 90$^{th}$ percentile station for one given country group falls within the green areas, this means that 90% of the available stations for that country fulfil the MQI criteria (see details in Thunis et al. (2013, 2013b)).

**5.2.1 NO$_2$**

With the exception of EMEP for Greece, the hourly MQI for the background NO$_2$ concentrations are satisfactorily modelled, regardless of the emission inventory. In general, the yearly MQI is more challenging to fulfil. The analysis of the following Figure 6 confirms this and all three models fail to fulfil the yearly MQI in most countries. From the country detailed information provided in the supplementary material, all models face important issues In Spain, Slovakia or Norway. In

general, CAMS shows the largest overestimation against measurements, especially in countries like Italy, Luxemburg or Slovenia. It is interesting to note that although no major difference is seen in terms of total NOx emissions in that country (Figure 3), the EMEP shows a very large overestimation in Greece that the other two models do not show. The number of countries for which both the hourly and yearly MQIs are fulfilled is 7, 11 and 12 for CAMS, EMEP and EDGAR, respectively.




**Figure 6: Target (left column) and scatter (right column) diagrams for NO₂ for the EU country groups. These two type of diagrams provide info on the daily maximum and yearly MQIs, respectively. Each point represents a country group that is located at a distance from the origin that corresponds to the MQI value of the 90th percentile worst station. In the target diagram, each point's ordinate corresponds to the mean country group normalized bias whereas its abscissa corresponds to the mean normalized CRMSE. Each point is located on the right or left side when correlation or standard deviation dominates the CRME error (see text for details). The green area represents the MQI fulfilment zone.**

### 5.2.2 Particulate matter (PM10 and PM2.5)

While modelled PM2.5 are generally in good agreement (Figure 7), this is not the case for PM10 (Figure 8) indicating an issue with the modelling of the coarse fraction of PM. Part of the worsening for PM10, compared to PM2.5, can be explained by the higher measurement uncertainty assumed for PM2.5 than for PM10 in the MQI Equations (2) and (3) therefore



allowing for less stringency on the model results when calculating the MQI for PM2.5. The EDGAR emission inventory leads to better performances for PM, especially for PM2.5, in the Eastern countries. The number of countries for which both the hourly and yearly MQIs are fulfilled is 22 for PM2.5 for all inventories whereas this number for PM10 drops to 3, 4 and 5 for the EMEP, CAMS and EDGAR models, respectively.


**Figure 7: Target (left column) and scatter (right column) diagrams for PM2.5 for the EU country groups. These two type of diagrams provide info on the daily and yearly MQIs, respectively. See additional explanations in caption of Figure 6.**



**Figure 8: Target (left column) and scatter (right column) diagrams for PM10 for the EU country groups. These two type of diagrams provide info on the daily and yearly MQIs, respectively. See additional explanations in caption of Figure 6.**

### 5.2.3 Ozone

For the summer 8h daily maximum $O_3$ concentrations, all models fulfil the 8h daily max MQI (target diagram) for the summer period, with the exception of the same three countries: Ireland, Malta and Romania. For the averaged concentrations (scatter plot), the models do not fulfil the MQI in three additional countries: Greece, Hungary and Norway. Regardless of the emission inventory use to feed the EMEP model, all models show a large overestimation with respect to the measured values, in countries characterised by low-ozone concentrations and show an underestimation in countries

where the measured ozone concentrations are the highest. This results in a "flat scatter" with modelled values that almost do not show any concentration difference among countries whereas the measured signal is strong. While spatial resolution is sometimes a limitation to the correct modelling of O₃, we believe that the current resolution of 7km should be sufficient to

capture spatial variations across the domain. Despite the fulfilment of the two MQIs in many countries, these results indicate the need for substantial improvements, other than emissions, before the model delivers results of a sufficient level to support policy applications for this pollutant.

**Figure 9: Target (left column) and scatter (right column) diagrams for O₃ for the EU country groups. These two type of diagrams**
**provide info on the 8h max daily and yearly MQIs, respectively. See additional explanations in caption of Figure 6.**





## 5.3. Evaluation at regional scale

In this section, we analyse the concentration modelled in the three regions introduced in section 3.4: The Benelux, the Po-valley and the Black triangle. Figure 10 to Figure 12 show the Target (for daily values) and scatter diagrams (for yearly/seasonal values) for PM2.5, $NO_2$ and $O_3$, respectively, while the detailed numbers for each group are summarized in Table 2. Regarding PM2.5, the Benelux (blue symbols) and the Po-valley region (orange symbols) are well modelled by the three models (all fulfil both the daily and yearly MQIs). Although EDGAR shows better performances with the fulfilment of the daily criteria over the Black Triangle region (Table 2), all models face issues in fulfilling the yearly MQI. In particular, models do not succeed to distinguish high and low concentration stations (all modelled values lie on the same horizontal line in the scatter diagram). For this region, the left/right position of each station in the target diagram informs on whether the model-observed discrepancy is dominated by correlation (i.e. the timing between model and measured peaks) or by the standard deviation (i.e. the compared amplitude of modelled and observed concentration variations). While for the Benelux and the Po valley, correlation is the main issue, standard deviation is the main issue for most stations located in the Black Triangle.



**Figure 10: Target (left column) and scatter (right column) diagrams for PM2.5 for all stations in the three selected regions (red: black triangle, orange: Po valley and blue: Benelux). These two type of diagrams provide info on the daily and yearly MQIs, respectively. See additional explanations in caption of Figure 6.**

In terms of NO₂ (Figure 11), all models fulfil the daily MQI but do fail to satisfy the yearly MQI. This is due to an overall model underestimation, especially in the Po-Valley and in the Black Triangle. This overall similar behaviour of the three models for that pollutant is coherent with the small differences observed in terms of NOx emissions in the three regions (Figure 4).

**Figure 11: Target (left column) and scatter (right column) diagrams for NO₂ for all stations in the three selected regions (red: black triangle, orange: Po valley and blue: Benelux). These two type of diagrams provide info on the hourly and yearly MQIs, respectively. See additional explanations in caption of Figure 6.**

As noted in the country analysis (Section 2), the EMEP model does not satisfactorily reproduce the $O_3$ concentrations (Figure 9), regardless of the emission inventory used to feed it. The model overestimate in low-level $O_3$ countries and underestimate in high-level $O_3$ countries, resulting in a "flat scatter" (points aligned along a horizontal line in the scatter plots). This behaviour is also visible in the three selected regions, with an overestimation in the Benelux and an underestimation in the Po-valley while the measured levels are best reproduced in the black triangle area. Again all models show very similar patterns despite the noted differences in terms of emissions, especially VOC (Figure 4). This seems to

indicate a low sensitivity of the model to VOC emissions. Note that these unsatisfactory results are in line with previous model evaluation (MACC, 2013), in terms of underestimation and flat signal.



**Figure 12: Target (left column) and scatter (right column) diagrams for O₃ for all stations in the three selected regions (red: black triangle, orange: Po valley and blue: Benelux). These two type of diagrams provide info on the 8h max daily and summer MQIs, respectively. See additional explanations in caption of Figure 6.**

Table 2 shows that the results obtained for PM10 (target and scatter not shown here for lack of space) are not as good as for PM2.5, especially over the Black triangle where the coarse fraction of PM is largely underestimated. Despite the much larger PM coarse (PMco) emissions included in EDGAR (factor 2 compared to EMEP) and the larger allocation of these emissions to urban areas (Figure 4), the impact on modelled PM10 concentrations remains limited.





**Table 2: MQI hourly/daily and yearly/summer MQI's Values for the three regions.**

|  | NO2 | | O3 | | PM2.5 | | PM10 | |
|---|---|---|---|---|---|---|---|---|
|  | **Daily** | **Yearly** | **Daily** | **Summer** | **Daily** | **Yearly** | **Daily** | **Yearly** |
| **CAMS** |  |  |  |  |  |  |  |  |
| Benelux | 0.68 | 1.37 | 0.54 | 0.42 | 0.62 | 0.37 | 0.86 | 1.10 |
| Black Triangle | 0.75 | 1.46 | 0.54 | 0.63 | 1.01 | 1.48 | 1.27 | 3.08 |
| Po-Valley | 0.89 | 1.86 | 0.69 | 0.99 | 0.93 | 0.70 | 1.16 | 1.99 |
| **GNFR** |  |  |  |  |  |  |  |  |
| Benelux | 0.60 | 1.18 | 0.54 | 0.42 | 0.63 | 0.36 | 0.86 | 1.02 |
| Black Triangle | 0.77 | 1.62 | 0.53 | 0.59 | 1.07 | 1.69 | 1.33 | 3.14 |
| Po-Valley | 0.88 | 1.81 | 0.70 | 0.97 | 0.93 | 0.75 | 1.17 | 1.97 |
| **EDGAR** |  |  |  |  |  |  |  |  |
| Benelux | 0.67 | 1.33 | 0.53 | 0.39 | 0.67 | 0.26 | 0.87 | 0.99 |
| Black Triangle | 0.76 | 1.48 | 0.51 | 0.48 | 0.97 | 1.29 | 1.23 | 2.84 |
| Po-Valley | 0.94 | 1.95 | 0.71 | 0.96 | 0.95 | 0.65 | 1.21 | 1.52 |

To assess how these emission-induced model differences reflect in terms of concentrations according to the station types, we
divided the available measurement background stations into three groups: urban, suburban and rural. For NO$_2$ (Figure 13, top
left), the EMEP inventory leads to slightly better results in the Benelux, especially for the urban background stations, while
results are very similar among models for the other two regions. For O$_3$ all model results are similar and lead to good results,
regardless of the station type.

For PM, EDGAR leads to improved results, especially in the Black Triangle and in a lesser measure in the Po Valley. The
improvement is larger in urban areas, probably because of the larger amount of total emissions as compared to the other
inventories but also because of the increased share of emissions allocated to the urban areas in this inventory. The larger
amount of SOx emissions in the EDGAR inventory might also play a role on the formation of secondary inorganic aerosols.



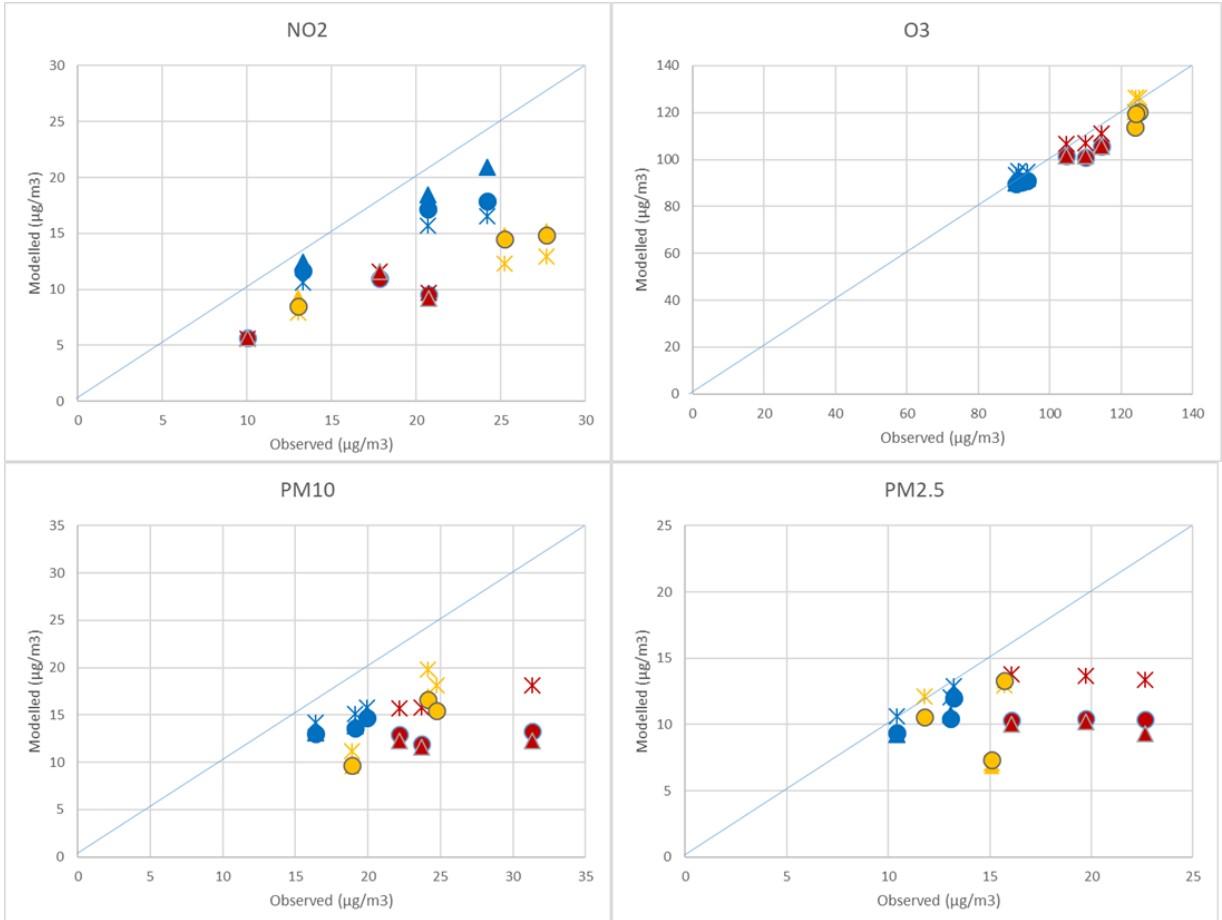

**Figure 13: Concentrations of NO₂, O₃ (MAX8H), PM10, PM2.5 and obtained with the three inventories for three types of stations. Regions are distinguished by different colours (as in Figure 10, 11 and 12) whereas symbols are used to differentiate the inventories. No distinction is made between urban, suburban and rural but these three groups are systematically found from right to left.**

## 6. Data availability

The EDGAR dataset (v5.0) used in this paper is available at: https://data.europa.eu/doi/10.2904/JRC_DATASET_EDGAR (link: EDGAR v5.0 Global Air Pollutant Emissions, Crippa et al., 2020a).

## 7. Discussion and conclusions

Given the regular updates and improvements made to the air quality models themselves, but also to their associated input data (in particular emissions), it is important to regularly perform sensitivity analysis to understand how these improvements impact the model results. However, emissions cannot be compared directly to a true value (as it does not exist) and the only available approach is indirect, i.e. compare measured and modelled concentrations obtained with different emission



inventories. Given the complexity of approach and because of technical requirements, additional limitations are present. For example, only one air quality model and one meteorological year could be tested in this study. Moreover, the comparison only addresses ground-level concentrations while some impacts would probably be noticed on vertical concentration profiles as well. Given the type of model (and spatial resolution) tested, the comparison is also limited to background stations.

Despite these limitations, we believe that some findings of this work can be useful to trigger further tests and improvements. One important issue is related to $O_3$ modelled patterns, which could be addressed by checking the behaviour of other models. While the important differences in terms of PM emissions observed in some regions (e.g. Black triangle) need to be discussed, they do not explain all of the large underestimation remaining in terms of concentrations. Therefore, either an important emission underestimation remains, either some processes is lacking within the model.

In this work, we applied the evaluation methodology proposed by FAIRMODE. It clearly illustrates the strengths and weaknesses of the modelling applications, in view of their use to support policy applications. As already said and clearly marked by the FAIRMODE evaluation methodology applied, Important progress remains necessary with regards to $O_3$ modelling, for which emission inventories do not seem to be the crucial lever to play with. Indeed, differences in NOx and particularly VOC emissions have a marginal impact on concentrations and spatial concentration gradients are missed by the

model. While the model behaves better for $NO_2$, issues remain the fulfilment of the yearly MQI. For PM, the situation is much better for PM2.5 than for PM10 for which an important underestimation persists. While these conclusions are general, the MQI based approach can also be used to distinguish the areas where a model behaves better than another one. The analysis performed over three regions (Po Valley, Black Triangle, and Benelux) shows that the emission differences and modelling issues are not geographically similar. The better results obtained over the Benelux, with regards to the other two

regions is a clear illustration of this.

It is important to stress the fact that most of the modelling issues raised above are not primarily depending on emissions. While this may be counter-intuitive as many publications point to emissions as the most uncertain model input and often as the key responsible for erratic modelled concentrations, we find in this work that important differences in terms of emissions do not always lead to large changes in terms of concentrations. Obviously, enlarging the tests to other models would allow

obtaining a more robust conclusion.

One of the main purposes of this work was also to assess how the EDGAR inventory compares to other reference inventories for air quality modelling. It is indeed one of the first applications of EDGAR to air quality modelling at European scale, as this inventory is primarily developed for air quality global simulations and green-house gas emission global estimation. The analyse shows that this inventory leads to results that are very comparable to the other two inventories, even leading to

improved results in some regions, especially in Eastern Europe. This is an important finding as the EDGAR inventory relies on an independent approach to estimate emissions, some of the differences being highlighted in this work.

Finally, to enrich the analysis performed in this manuscript, it would be useful to collect also bottom-up (local) emission inventories, that could provide a useful benchmark to be used to evaluate the quality of EU-wide emission inventories. As a

final product, in this way we would improve EU-wide emission inventories, to be used to design policies and evaluate their
impact on air quality.

**Author contributions.**

PT and CC did the majority of the analysis in the paper, and wrote main part of the results on the concentration side. MC,
DG and GO were in charge of the development of the EDGAR emission inventory and of the comments on the emission part
of the manuscript. ADM run the simulations with the EMEP model and performed initial analysis on the results. EP had the
initial idea for the paper, did the first analysis and coordinated the paper writing.

**Competing interests.**

The authors declare that they have no competing interests.

**Acknowledgements.**

We acknowledge both the EMEP and CAMS community, that provide the open source EMEP air quality model, and the
EMEP and CAMS emission inventories.

**Supplementary material**
The Figures in this section provides information detailed at country level on the hourly/daily and summer/yearly Model
Quality Objective (MQI) for the three models.

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
