# Peer review of "Sensitivity of air quality modelling to different emission inventories: a case study over Europe"

_Earth System Science Data, 2020_

## Referee Comment (RC1) · Terry Keating (Referee) · 21 Aug 2020

This paper presents the EDGARv5 emissions inventory and evaluates its use in regional air quality modeling for Europe, in comparison with two other available emissions inventories. In addition to presenting the data set, the paper uses the FAIRMODE model evaluation methods to explore the implications of using the different emissions inventories. The analysis presented is useful and significant. The data set is of high quality and well documented. The presentation is good, but warrants some editting. In particular, the conclusions section would be stronger if it were better organized. Some explanations are not clear and are discussed in the line by line comments below.

Lines 33-34: The opening sentence of this paragraph contains several problems. A bet-

[Figure]

[Figure]

ter formulation would be: "Numerous studies have assessed the sensitivity of modelled concentrations and related indicators to the choice of emissions inventory or emissions uncertainty."

Lines 38-39: "with an uncertainty (due to emissions variability)" confuses the concepts of uncertainty and variability. This needs to be rephrased.

Lines 44-49: "used in the EMEP model for their policy applications" is a problematic phrase (Whose policy applications? Should this refer to the Convention on Long-range Transboundary Air Pollution?). Why not just provide the labels and refer to Table 1? Why do you refer to CAMS here and later CAMS-REG-AP? This should be consistent and better to refer to CAMS-REG-AP for the emissions data set and CAMS for the service overall.

Line 49: Should "In Section 2, . . ." begin a new paragraph?

Line 69: Here you refer to CAMS-REG-AP instead of CAMS. Need to be consistent, and add REG-AP designation to previous section (as noted above).

Line 71: it is not necessary to have the phrase "on the contrary," and it is distracting.

Line 132-4: this could be stated more clearly: "the use of EMEP as a reference point has no implication on the quality of the inventory itself, . . ."

Line 159: I'm not sure that this is obvious, at least as presented. In lines 148-159, I believe that the correlations being plotted are the fraction of EU emissions that are assigned to each country. This should be made more clear. Then, yes the fraction of EU emissions in each country is correlated with population. Thus, the correlations between inventories are influenced by this relationship.

Line 162-164: It would be useful to explain the significance of the differences increasing when the comparison is made on a per capita basis. By controlling for population (i.e., the activity level), you see greater differences in the assumptions about technology (i.e., the emission factors).

Line 175-179: This bullet needs to be edited for language.

Line 182-190: This section is not presented clearly. Total European SOX emissions are higher in EDGAR than in the other inventories. (Is this the difference observed in Eastern Europe?) However, the explanation provided by the authors does not explain the overestimation. EDGAR assumes all plants respect the 2001 LCP Directive. (The fact that the LCP Directive was updated in 2016 does not play into this case as we are only looking at 2015 emissions.) If small plants are not required to meet the same emission rate limits as large plants, then the EDGAR methodology will underestimate emissions from small plants and the overall sector. The authors also speculate that the UDI Platts database may overestimate how much energy is produced in small plants. Given EDGAR assumes all plants comply with the LCP limits, it is not clear that this has any impact. However, if the emission factors from small plants were assumed to be higher than for large plants, then this overestimate of small capacity might explain a higher total for EDGAR than for the other inventories.

Line 234: Figure 4. I think that the display of the ratio of total emissions by bars and urban emissions by circles works well. The caption should refer to the circles as does the text, not "bullets."

Line 239: "spatial yearly mean concentration fields" seems repetitive. Better to use "annual mean concentration field."

Line 242: "but they are also widespread..." introduces a vague "they". It would be better to replace with "but also ..."

Line 253: Is this model validation or model evaluation?

Line 379: Is the phrase "for this region" correct?

Line 399: Comparing Figure 12 to Figure 9, it appears that the models perform much better in the selected regions than across the whole domain. This does not seem to be reflected here in the text.

Table 2: Is GNFR supposed to be EMEP? The caption should explain that the numbers in green are below 1 and represent agreement of the model and observations within 2x the observational uncertainty.

Line 417: The conclusion that O3 model results are good contradicts the text at line 399. It would be useful to explain why the regional performance may be better than the domain wide performance, which is especially poor for areas with low observed concentrations.

Line 440: From this line on, the conclusions bounce around a bit from O3, to PM, to methods, back to O3, PM, NOx... A more systematic reckoning would be easier to follow: What did we learn about NOx? What did we learn about O3? What did we learn about PM?

Line 445: I suggest moving the mention of the FAIRMODE methodology to the section before line 440, i.e., before you start getting into the implications for O3 and PM modeling.

Some important conclusions: 1) Differences in emissions do not account for poor model performance for PM (underestimation) and O3 (overestimation). a) For O3, differences in NOx and VOC emissions have marginal impacts on concentrations and spatial gradients are missed by the model. b) For PM, performance is better for PM2.5 than PM10, which is underestimated. This suggests a missing emissions source or modeled process. 2) EDGAR compares well to other inventories for air quality modeling.

Line 447: Important should not be capitalized.

Line 464: Suggest "analysis" as opposed to "analyse"

---

## Author Comment (AC1) · 11 Sep 2020

This paper presents the EDGARv5 emissions inventory and evaluates its use in regional air quality modelling for Europe, in comparison with two other available emissions inventories. In addition to presenting the data set, the paper uses the FAIRMODE model evaluation methods to explore the implications of using the different emissions inventories. The analysis presented is useful and significant. The data set is of high quality and well documented. The presentation is good, but warrants some editing. In particular, the conclusions section would be stronger if it were better organized. Some explanations are not clear and are discussed in the line by line comments below.

Lines 33-34: The opening sentence of this paragraph contains several problems. A better formulation would be: "Numerous studies have assessed the sensitivity of modelled concentrations and related indicators to the choice of emissions inventory or emissions uncertainty."

*We thank the reviewer for the comments. We updated this part of the manuscript as suggested.*

Lines 38-39: "with an uncertainty (due to emissions variability)" confuses the concepts of uncertainty and variability. This needs to be rephrased.

*The text will be rephrased as follows: "They estimated 2.1 million premature deaths per year due to $PM_{2.5}$ concentrations, with an uncertainty (due to emissions uncertainties) of more than 1 million premature deaths per year.*

Lines 44-49: "used in the EMEP model for their policy applications" is a problematic phrase (Whose policy applications? Should this refer to the Convention on Long-range Transboundary Air Pollution?). Why not just provide the labels and refer to Table 1? Why do you refer to CAMS here and later CAMS-REG-AP? This should be consistent and better to refer to CAMS-REG-AP for the emissions data set and CAMS for the service overall.

*'CAMS' has been replaced with 'CAMS-REG-AP' in all instances, when referring to the emissions data. To clarify 'EMEP', we modified the sentence as follows: "We use here (and in the rest of the paper) the 'EMEP' label to refer to the emissions as used in the EMEP model in the context of "the Convention on Long-range Transboundary Air Pollution" (EMEP, 2019; see https://www.ceip.at/ms/ceip_home1/ceip_home/webdab_emepdatabase/emissions_emepmodels/)."*

Line 49: Should "In Section 2, : : :" begin a new paragraph?
*We modified the text as suggested.*

Line 69: Here you refer to CAMS-REG-AP instead of CAMS. Need to be consistent, and add REG-AP designation to previous section (as noted above).
*As said previously, we modified all references to CAMS to CAMS-REG-AP.*

Line 71: it is not necessary to have the phrase "on the contrary," and it is distracting.
*We modified the text as suggested.*

Line 132-4: this could be stated more clearly: "the use of EMEP as a reference point has no implication on the quality of the inventory itself:"
*We modified the text as suggested.*

Line 159: I'm not sure that this is obvious, at least as presented. In lines 148-159, I believe that the correlations being plotted are the fraction of EU emissions that are assigned to each country. This should be made more clear. Then, yes the fraction of EU emissions in each country is correlated with population. Thus, the correlations between inventories are influenced by this relationship.
*Right, we modified the sentence as follows: "For this comparison, emissions have been summed up sector-wise, to evaluate the level of similarity between the country repartition of the EU overall emissions for each pollutant."*

Line 162-164: It would be useful to explain the significance of the differences increasing when the comparison is made on a per capita basis. By controlling for population (i.e., the activity level), you see greater differences in the assumptions about technology (i.e., the emission factors).
*We added the following text: "To prevent this problem, we provide the same emission comparison, but per capita (Figure 2 bottom). Correlations drop significantly with the exception of NH3 and PM2.5. With this per-capita normalisation, the implicit correlation between emission and population (i.e., the activity level) is removed and the main observed differences are then related to varying assumptions on technology (i.e., the emission factors).*

Line 175-179: This bullet needs to be edited for language.
*We rewrote the bullet points as follows:*
- *All inventories agree for NOx and PM2.5 emissions, for which the bias is small (Figure 3 top-right);*
- *For the coarse fraction of PM (Figure 3 bottom-right), the CAMS-REG-AP and EMEP inventories agree, but differ from the EDGAR estimates, that shows larger emission in several countries. This difference is mostly occurring in Eastern countries, but countries such as Italy or Spain are also concerned;*
- *Differences for SOx emissions (mainly generated by power and heat plants, especially fuelled with coal) between EDGAR and other inventories are also observed, especially in Eastern European countries. A possible reason for this discrepancy is due to the fact that EDGAR SOx emissions for this sector are calculated using the capacity dependant limits existing in 2001/2002. In fact, for each country, national implied emission factors, using the capacity for each fuel type and technology, were calculated using the data in UDI Platts database, considering emission limits existing in 2015 (following 2001/2002 regulations). However, in reality, as in 2016 new emission regulations were going to be in force, probably industries were already in a transition phase to reduce their emissions, and this may not have been properly captured in our estimation methodology.*
- *For VOC (Figure 3 bottom-left), the EDGAR overestimation (with respect to EMEP) is the largest in countries like Belgium, Austria, Switzerland, Germany or Finland, with differences mainly in industrial combustion and fugitive emissions sectors.*

Line 182-190: This section is not presented clearly. Total European SOX emissions are higher in EDGAR than in the other inventories. (Is this the difference observed in Eastern Europe?) However, the explanation provided by the authors does not explain the overestimation. EDGAR assumes all plants respect the 2001 LCP Directive. (The fact that the LCP Directive was updated in 2016 does not play into this case as we are only looking at 2015 emissions.) If small plants are not required to meet the same emission rate limits as large plants, then the EDGAR methodology will underestimate emissions from small plants and the overall sector. The authors also speculate that the UDI Platts database may overestimate how much energy is produced in small plants. Given EDGAR assumes all plants comply with the LCP limits, it is not clear that this has any impact. However, if the emission factors from small plants were assumed to be higher than for large plants, then this overestimate of small capacity might explain a higher total for EDGAR than for the other inventories.
*See the previous response, in which we tried to clarify this issue.*

Line 234: Figure 4. I think that the display of the ratio of total emissions by bars and urban emissions by circles works well. The caption should refer to the circles as does the text, not "bullets."
*We corrected as suggested.*

Line 239: "spatial yearly mean concentration fields" seems repetitive. Better to use "annual mean concentration field."
Line 242: "but they are also widespread: : :" introduces a vague "they". It would be better to replace with "but also : : :"
*We corrected as suggested.*

Line 253: Is this model validation or model evaluation?
*It is evaluation. We corrected.*

Line 379: Is the phrase "for this region" correct?
*Yes, it is correct.*

Line 399: Comparing Figure 12 to Figure 9, it appears that the models perform much better in the selected regions than across the whole domain. This does not seem to be reflected here in the text.
*The difference is linked to the fact that in Figure 9 results are shown as aggregated per country, while Figure 12 shows single station results. This is explained in the text and captions.*

Table 2: Is GNFR supposed to be EMEP? The caption should explain that the numbers in green are below 1 and represent agreement of the model and observations within 2x the observational uncertainty.
*We thank the reviewer for the comments. It is a typo. We now replaced GNFR with EMEP, in Table 2. We also updated the caption, as follows: "MQI hourly/daily and yearly/summer MQI's Values for the three regions. The numbers in green in the table are below 1, and represent a good agreement between model and observations as defined by the Model Quality Indicator."*

Line 417: The conclusion that O3 model results are good contradicts the text at line 399. It would be useful to explain why the regional performance may be better than the domain wide performance, which is especially poor for areas with low observed concentrations.
*This behaviour has been better explained with this sentence: "In contrast to the countries 'grouped' validation presented in Section 5.2, for the selected three regional domains all model results are similar for O3, leading to good results regardless of the station type."*

Line 440: From this line on, the conclusions bounce around a bit from O3, to PM, to methods, back to O3, PM, NOx: : : A more systematic reckoning would be easier to follow: What did we learn about NOx? What did we learn about O3? What did we learn about PM?
Line 445: I suggest moving the mention of the FAIRMODE methodology to the section before line 440, i.e., before you start getting into the implications for O3 and PM modeling.
Some important conclusions: 1) Differences in emissions do not account for poor model performance for PM (underestimation) and O3 (overestimation). a) For O3, differences in NOx and VOC emissions have marginal impacts on concentrations and spatial gradients are missed by the model. b) For PM, performance is better for PM2.5 than PM10, which is underestimated. This suggests a missing emissions source or modeled process. 2) EDGAR compares well to other inventories for air quality modeling.
*The conclusions have been reorganised and re-phrased, following suggestions from the Reviewer*

Line 447: Important should not be capitalized.
Line 464: Suggest "analysis" as opposed to "analyse"

*Fixed.*

*Fixed.*

---

## Referee Comment (RC2) · Anonymous Referee #2 · 30 Sep 2020

The article presents a comparison of chemistry-transport model simulation using the EMEP model over Europe based on three different sources of anthropogenic emissions (EMEP, CAMS-REG, and EDGAR). It addresses the interesting question of model sensitivity to air pollutant emission fluxes.

I would however not recommend its publication in ESSD. First because the paper is essentially a model evaluation work, therefore more suited to a modelling journal such as Copernicus' Geos. Model Dev. In addition, the paper does not support the publication of any geophysical dataset: EDGAR is being published elsewhere (Crippa et al. 2018 is a good example of an article supporting dataset publication) and too poorly documented here to legitimate an update.

I also have the following other concerns:

- Why is EDGAR presented with more details than CAMS-REG or EMEP emission in the paper?

- The background literature is very weak in the introduction, there are more articles of relevance with the present study than Zhu et al. 2019

- Is the same LRTAP reporting year used in the EMEP and CAMS-REG emissions for 2015? Presumably not if the paper states that EMEP is available until 2017 and CAMS-REG until 2016. Can there be any influence on the national emission comparisons?. There are also inaccuracies in the data description as CAMS-REG v2.2.1 was only extending up to 2015.

- L74: what is less transparent in EMEP & CAMS-REG inventories than in EDGAR?

- L131: why not using EDGAR as reference given the stronger focus given to that emission source?

- L154: from the figure I would say that PMco inconsistencies originate from industry and traffic

- L156: how can these differences on shipping and aviation explain the discrepancy on industrial emission mentioned in the previous sentence?

- L160 to 200: and following: the comparison between EMEP/CAMS and EDGAR is not uninteresting, but also largely expected since EMEP&CAMS are based on similar national emissions (when using identical reporting years).

- L299 (and following): the three different model setup can be referred as "three simu-lations" but certainly not as "three models"

- L331: it is worth pointing out this difference in NO2 performance despite similar NOx emission, but it would be interesting to explain why

- L344: why stating here that the different model quality is related to an issue in the model itself whereas the following sentence actually point towards the sensitivity to

observation uncertainty in the MQI design?

- L366: as pointed out by the authors, the poor performances for ozone is a concern. Also because the EMEP model is widely used for policy support, especially for ozone. I could not find the MACC-III Report reference (2013), but this feature should be further explained, for instance by checking biogenic emissions. This point again legitimates my major comment about the relevance to submit the present paper to a modelling journal.

- L418: is such a good performance for ozone consistent with earlier statements?